# Application of Tungsten-Oxide-Based Electrochromic Devices for Supercapacitors

Muyun Li [1], Haoyang Yan [1], Honglong Ning [1,*], Xinglin Li [1], Jinyao Zhong [1], Xiao Fu [1], Tian Qiu [2], Dongxiang Luo [3], Rihui Yao [1,*] and Junbiao Peng [1]

[1] State Key Laboratory of Luminescent Materials and Devices, South China University of Technology, Guangzhou 510640, China; llmyscut@163.com (M.L.); 201930177088@mail.scut.edu.cn (H.Y.); 202121023273@mail.scut.edu.cn (X.L.); 202010103138@mail.scut.edu.cn (J.Z.); 201630343721@mail.scut.edu.cn (X.F.); psjbpeng@scut.edu.cn (J.P.)
[2] Department of Intelligent Manufacturing, Wuyi University, Jiangmen 529020, China; qiutian@ustc.edu
[3] Guangzhou Key Laboratory for Clean Energy and Materials, Huangpu Hydrogen Innovation Center, Institute of Clean Energy and Materials, School of Chemistry and Chemical Engineering, Guangzhou University, Guangzhou 510006, China; luodx@gdut.edu.cn
* Correspondence: ninghl@scut.edu.cn (H.N.); yaorihui@scut.edu.cn (R.Y.)

**Abstract:** For making full use of the discoloration function of electrochromic (EC) devices and better show the charge and discharge states of supercapacitors (SCs), electrochromic supercapacitors (ECSCs) have attracted much attention and expectations in recent years. The research progress of tungsten-oxide-based electrochromic supercapacitors (ECSCs) in recent years is reviewed in this paper. Nanostructured tungsten oxide is widely used to facilitate ion implantation/extraction and increase the porosity of the electrode. The low-dimensional nanostructured tungsten oxide was compared in four respects: material scale, electrode life, coloring efficiency, and specific capacitance. Due to the mechanics and ductility of nano-tungsten oxide electrodes, they are very suitable for the preparation of flexible ECSCs. With the application of an organic protective layer and metal nanowire conductive electrode, the device has higher coloring efficiency and a lower activation voltage. Finally, this paper indicates that in the future, $WO_3$-based ECSCs will develop in the direction of self-supporting power supply to meet the needs of use.

**Keywords:** electrochromic supercapacitors; tungsten oxide; low-dimensional nanostructure; self-supporting power supply

## 1. Introduction

With the continued interest in environmental protection and energy conservation, electrochromic technology with low voltage drive and dynamic control of solar heat and light input to buildings is developing rapidly as a very promising energy-saving technology [1–7]. Products such as smart windows, anti-glare mirrors, and aircraft portholes all involve electrochromic technology, which is already common in our daily life [3,8,9]. According to a study by David R. Roberts of the US Renewable Energy Laboratory, residential homes with electrochromic smart windows save 9.1% in total energy consumption and 13.5% in electricity consumption compared to homes with low-e glass and shaded glass [10].

Amorphous tungsten oxide is a widely investigated cathode EC material that has high coloring efficiency, color switching response speed, and stability in the process of electrochromism. Compared with other EC materials, $WO_3$ does not undergo a phase change during electrochromic coloration and electrochemical reaction, thus showing a good rate capability [11–20]. Deb [21] first reported that $WO_3$ can display transparent and blue colors by alternating the application of positive and negative voltages. The $WO_3$ structure, which strictly satisfies the stoichiometric ratio, can be seen as an $ABO_3$-type chalcogenide structure with the A atom absent. The W and O are mainly linked by ionic

bonds, with a few covalent bonding components, a large number of dangling bonds, and defects all over the $WO_3$ structure, which are very favorable for electrons to enter the crystal to accelerate the reaction. The electrochromic reaction of tungsten oxide follows the following chemical formula:

$$WO_3(\text{bleached}) + xM^+ + xe^- \leftrightarrow M_xWO_3(\text{coloured})$$

where $M^+$ is $H^+$, $Li^+$, $K^+$, $Na^+$, etc., and the color change is achieved by reversible ion injection/extraction in the $WO_3$ film. Figure 1 shows a diagram of the continuous electron-transfer process in electrochromism, which can be described as two steps:

$$W^{6+} + e^{-1} \leftrightarrow W^{5+}$$
$$W^{5+} + e^{-1} \leftrightarrow W^{4+}$$

Tungsten oxide has a high intrinsic density with a rich skeletal diversity, and is therefore an extremely promising material for energy-storage electrodes [22–26]. Tungsten oxide with different morphologies can be obtained via magnetron sputtering, as well as by hydrothermal, electrochemical, or vapor phase deposition [27–29].

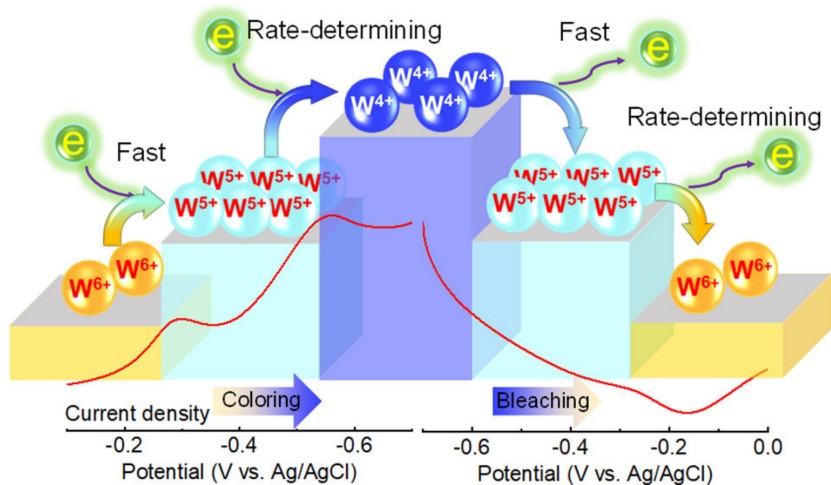

**Figure 1.** Diagram showing the rate-determining step in the continuous electron-transfer process [9]. [Reprinted with permission from Ref. [9] Copyright {2021} American Chemical Society.]

General Electric patented the first supercapacitor in 1958, which used activated carbon as a substrate, where charge was not exchanged between the electrode and the electrolyte, and where the electrode simply stored charge. Carbon materials and their derivatives, which have a large specific surface area and good electrical conductivity, therefore have a wide range of applications in supercapacitors [29–33]. The output power versus stored charge is significant. According to Figure 2, output power and stored energy density are important parameters of energy storage devices, which can be classified as capacitors, supercapacitors, batteries, fuel cells, etc. [1,8,33–36]. Compared to batteries, which have a higher energy density but lower transmission density, supercapacitors have a lower energy density but higher transmission density, which comes from the large specific surface area of their electrodes but lower densification. Electrochromic devices have many similarities to supercapacitors and, therefore, have the potential to be integrated. Structurally, both are sandwich structures consisting of an electrode layer and an electrolyte layer, as shown in Figure 3. In terms of operating principles, both electrochromic devices and pseudocapacitors are based on redox reactions for energy transfer and storage [36–39]. In terms of parameters, both supercapacitors and electrochromic devices have fast switching times and low operating voltages. The electrode is the core component of a capacitor, and its performance is influenced by its strengths and weaknesses. Generally speaking, the following requirements are imposed on electrodes: (1) large ion embedding and stripping (i.e., high specific capacity); (2) good reversibility of embedding and stripping, small structural

changes (i.e., long cycle life); (3) high ion diffusion coefficient and electron conductivity (i.e., low temperature, good multiplication characteristics); (4) high chemical/thermal stability, good compatibility with electrolyte (i.e., good safety); (5) abundant resources, environmental friendliness, and low price (i.e., low cost, environmental protection) [40–44]. Therefore, by designing the structure of the electrochromic material ($WO_3$) to meet the electrode requirements of the supercapacitor, the electrochromic supercapacitor can react to the charging and discharging state of the device through color change (the electrolyte layer is not mentioned separately because it has been introduced in considerable detail in the field of energy [45–47]).

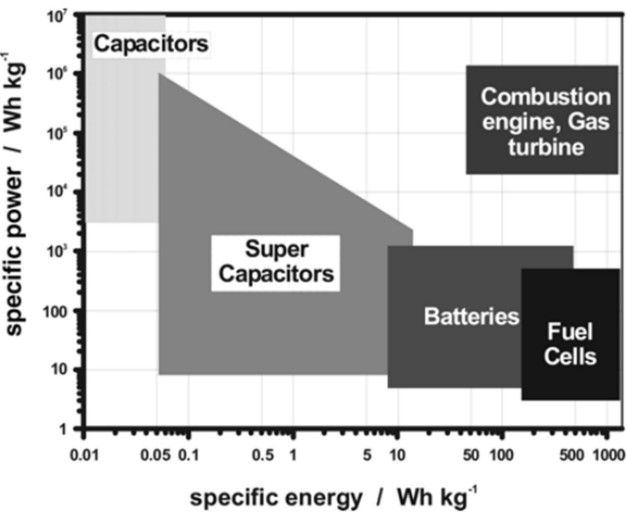

**Figure 2.** Ragone plot of different electrochemical energy conversion systems, combustion engines, turbines, and traditional capacitors [48]. [Reprinted with permission from Ref. [48] Copyright {2004} American Chemical Society.]

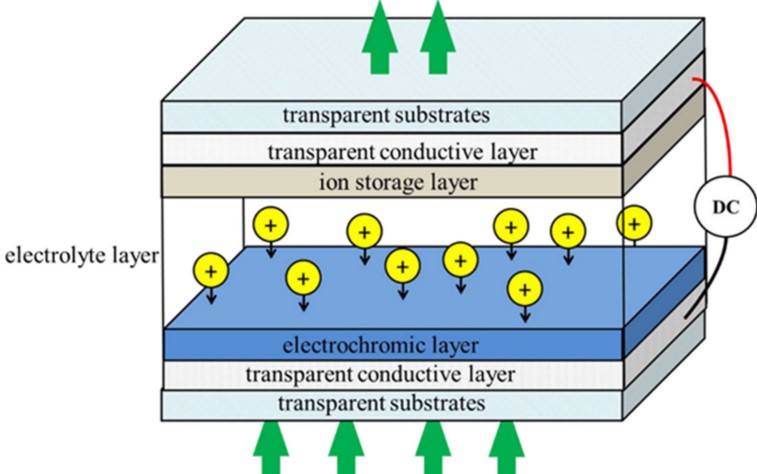

**Figure 3.** A typical ECD design. Arrows indicate the movement of ions and electrons in an applied electric field [4].

An electrochromic supercapacitor is different from an electrical double-layer capacitor; it uses fast and reversible surface redox reactions, and its capacitance is not limited to static electricity, but also includes electrochemical charge transfer, so it is called a pseudo-capacitor [49–53] (pseudo is used to distinguish electrostatic capacitors). Because of the more complex redox reactions, pseudocapacitors usually have a greater specific capacitance than carbon double-layer capacitors, and a large number of metal oxides are used in this field [54,55]. The energy (E) stored in the capacitor is $E = 1/2CV^2$; therefore, the research

focuses on how to increase the specific capacitance and voltage window of the device while ensuring a certain level of electrochromism, the key to the former being to obtain electrodes with a high specific surface area, and the key to the latter being to improve the overall environment of the device. In this review, we introduce the latest research progress of tungsten-oxide-based electrochromic devices in supercapacitors in recent years. This includes nanometer tungsten oxide electrodes, transparent conductive electrodes, and some multifunctional ECSC integrated devices.

## 2. Tungsten Oxide Electrodes

Liang [56] previously mentioned that tungsten oxide has the potential to combine electrochromism and energy storage, and in 2014, Yang [57] deposited $WO_3$ on FTO glass and prepared electrochromic supercapacitors with specific capacitance at 639.8 F/g and an optical modulation amplitude of 76%. The size of the synthesized polycrystalline $WO_3$ was above 100 nm, and the voltage driving the electrochromism was greater than the capacitance and the lifetime test voltage of the capacitor at 3.0 V, which may pose a hidden problem for the use of composite devices. In recent years, tungsten oxide nanostructures have been extensively researched, and the nanostructured electrochromic materials have a very large specific surface area, which more obviously reduces the driving voltage of the device and makes it more compatible with the driving voltage of supercapacitors [19–21,58–62]. In addition, tungsten oxide nanostructures are suitable for most processing methods, such as magnetron sputtering, spray-film printing, screen printing, spraying, and roll-to-roll processes [63–68]. The porosity of tungsten oxide nanoelectrodes is also suitable for most processing methods, such as magnetron sputtering, film printing, screen printing, coating, and roll-to-roll processes. The porosity of tungsten oxide nanoparticles is extremely high, but many problems arise, such as agglomeration of the material, degradation of the discoloration properties, and collapse of the material structure during operation. Therefore, in order to ensure a high specific capacitance and a long life, tungsten oxide nanoparticles are often used in combination with other organic and inorganic materials.

### 2.1. Zero Dimensions

The zero-dimensional (0D) material refers to materials with three dimensions in the nanoscale range (0–100 nm), or consisting of them as the basic unit. Due to the significant proportion of surface atoms, their surface density of states is greatly increased at the same time; for such small particles, certain quantum effects (e.g., quantum size effect, quantum limitation effect, quantum tunneling effect, quantum interference effect, etc.) are notable. The small size of 0D materials in the electrochromic device significantly shortens the diffusion path of the intercalated ions in the solid phase, (1)which helps to achieve fast charge/mass transfer; (2) the large surface-to-volume ratio of the quantum dots facilitates close contact between the electrode material and the electrolyte/collector, thus providing fast charge-transfer and electron-transfer kinetics; and (3) the significantly higher ratio of surface atoms makes the quantum dot electrodes more active in electrochemical reactions. Zhao [69] reported tungsten oxide quantum dot electrochromic supercapacitors, where tungsten oxide quantum dots of 3 nm in size were synthesized by a hydrothermal method, as shown in Figure 4. Compared to bulk tungsten oxide, the quantum dot tungsten oxide has a wider forbidden band, and the Faraday electrochemical process is accelerated, showing excellent electrochemical and electrochromic behavior. Another major task of the paper was to maintain the dispersion stability of the quantum dots, which were wrapped in an octane diamine ligand and modified with a conductive pyridine ligand on the surface. The final device had a fading time of 1 s and a coloring efficiency of 154 $cm^2$/C. The synthesis of tungsten oxide quantum dots by hydrothermal methods is difficult, mainly because the material tends to agglomerate into clusters or lumps during the synthesis process. However, in the synthesis of other dimensional materials, low concentrations of tungsten oxide quantum dot byproducts can be obtained, which can be purified to obtain a high-concentration solution [70].

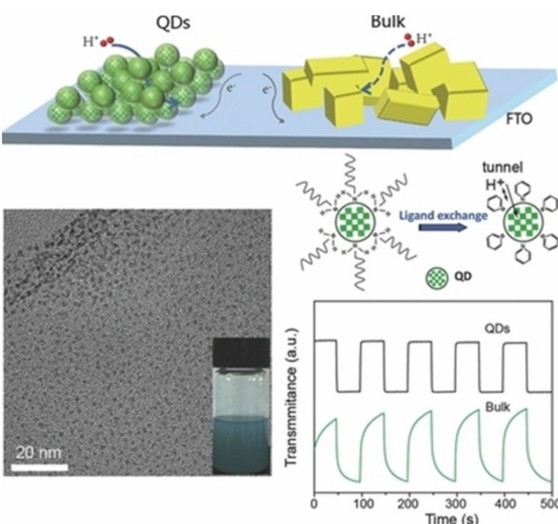

**Figure 4.** Comparison of bulk (green) and quantum dot (black) tungsten materials [66].

Inamdar [71] prepared oxygen-enriched nanometer tungsten oxide by adjusting magnetron sputtering parameters. The particle size of tungsten oxide was less than 10 nm, as shown in Figure 5, and a large voltage window (1.4 V) was obtained, with a specific capacitance of 228 F/g at 0.25 A/g, maintaining 75% of the initial capacitance for 2000 cycles, and a color rendering efficiency of 170 $cm^2$/C. It was verified that oxygen-rich tungsten oxide can accelerate the modulation speed of the device, because the excess oxygen element introduces more defects into the microstructure and achieves the purpose of containing more Li+ ions. This is also similar to the reason why amorphous tungsten oxide films have faster response speed than crystalline tungsten oxide films, both of which increase the porosity of the films [72].

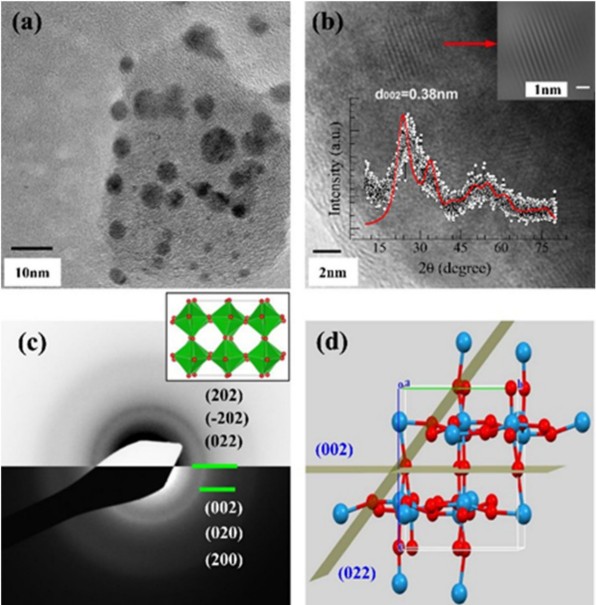

**Figure 5.** (**a**) TEM and (**b**) HR-TEM images with 10 and 2 nm scale bars, respectively. The inset shows the X-ray nature of the synthesized film. The SAED pattern of the corresponding film is shown in (**c**), and the inset shows that the monoclinic $WO_3$ phase is mostly composed of corner-linked octahedra of 4-membered channels along all crystallographic directions. The 002 and 022 crystallographic planes obtained from the SAED patterns are projected in (**d**). All crystallographic projections were made with the VESTA structure drawing software [71].

There is little research on the working mechanism of 0D tungsten oxide particles, and the quantum effect of tungsten oxide in quantum dots has not been observed in the discoloration process, while the synthesis of nano-tungsten oxide particles is also difficult, and the life of the electrode is short. However, its coloring efficiency and specific capacitance are outstanding. If the preparation process is further improved, the difficulty of preparation is reduced, and the life of the electrode is extended, the application prospects of 0D tungsten oxide materials will be more extensive.

### 2.2. One Dimension

The one-dimensional (1D) tungsten oxide nanostructure is one of the most widely used WO$_3$ nanostructures. Nanowires, nanorods, and nanotubes are all kinds of one-dimensional materials, as shown in Figure 6. Hydrothermal, electrochemical, vapor deposition, and electrostatic spinning methods can all be used to synthesize 1D tungsten oxide nanostructures [73–85], and the hydrothermal method is the most used and the simplest one. The synthesis mechanism is that during the formation of tungsten oxide, the guiding agent is used to wrap the material to make the material grow in one direction, and nanostructures with different aspect ratios can be obtained by controlling the temperature, time, and material ratio.

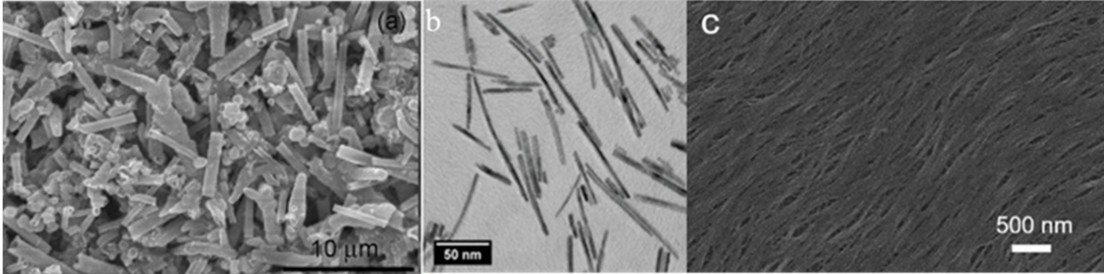

**Figure 6.** Different one-dimensional tungsten oxide nanostructures: (**a**) nanotubes [73], (**b**) nanorods [74], (**c**) nanowires [75].

Reddy [86] prepared WO$_3$ fibers with lengths of 50–200 nm and widths of 20–60 nm by a hydrothermal method, and deposited them electrophoretically on a conductive substrate. Au nanoparticles and PEDOP were coated on the surface of WO$_3$ to obtain PEDOP–Au@WO$_3$ composite electrodes, and it was shown that this hybrid electrode can effectively replace the expensive Pt-based counter-electrode solar cells, while the one-dimensional tungsten oxide material can form a functional framework on the flexible substrate at low temperatures or even room temperature, so flexible electrochromic device supercapacitors are an inevitable direction of development [7]. Li [84] prepared composite membranes with graded porous structures via vacuum-assisted filtration of mixed dispersions containing W$_{18}$O$_{49}$ nanowires and single-walled carbon nanotubes, and assembled a flexible, asymmetric electrochemical capacitor based on aluminum ions. The high degree of connectivity of nanowires and nanotubes formed an interpenetrating nanonetwork that ensured fast ion transport and high conductivity of the electrodes (1626 S/cm). The flexible devices need to be fabricated according to the suitability of thin films and flexible substrates, and after fabrication, their use under bending and stretching should be considered. Yun [85] found that the stability of the ECD supercapacitor was further improved on the basis of the flexible and stretchable WO$_3$ nanotubes combined with PEDOT:PSS in a double stack, as shown in Figure 7 The WO$_3$ nanotubes provided a very high specific capacitance and electrochromic properties, while the PEDOT:PSS stacked on top improved the mechanical properties of the device. The final device had a maximum specific capacitance of 471.0 F/g, a capacity retention of 92.9% after 50,000 charge/discharge cycles, and a color rendering efficiency of 83.9 cm$^2$/C. The use of tungsten oxide nanostructures in the paper effectively improved the film's mechanical properties, the use of PEDOT:PSS improved the compatibility of the film with the flexible substrate, and the use of gel electrolytes effectively solved

the impact of liquid leakage on the lifetime of the device. One-dimensional tungsten oxide electrodes are characterized by high density of stacked materials, but the stacked materials lead to more nodes and uneven impedance distribution, so it is important to order the 1D structure. $WO_3$–Ag nanowire lattices were constructed using the L-B method. The $WO_3$ lattices were arranged in an orderly fashion on a flexible substrate, and this method of large-area co-assembly of nanowires can be used for a variety of flexible nanowire devices [86]. ECSCs fabricated on flexible substrates have great application prospects, while ECSCs on rigid substrates can obtain very high device parameters [87–93]. Prasad [94] mixed tungsten oxide ($WO_3$) and vanadium oxide ($V_2O_5$) to prepare an ECSC. The optical contrast of the device was 60%, the fast color response was 4.9 s, and the highest color efficiency was 61.5 $cm^2$/C. At a current of 0.5 $mA/cm^2$, the maximum area capacitance of 38.75 $mF/cm^2$ was obtained. In addition, even after 5000 charge/discharge cycles, the capacitance retention was still 78.5%. The unique morphology of nanostructures, the feasible redox reaction caused by the existence of active sites, and the high charge-transfer rate are the main factors to improve the electrochromism and electrochemical energy storage properties [95–97].

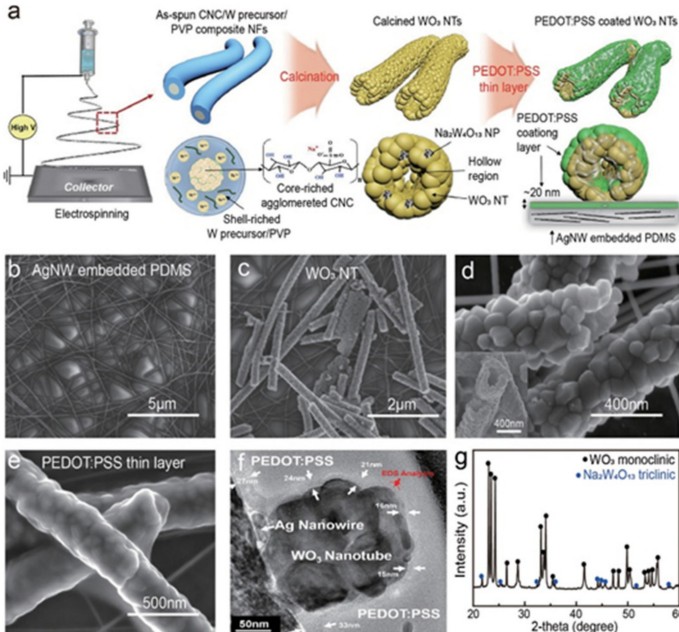

**Figure 7.** (**a**) Schematic illustration of the fabrication process of $WO_3$ nanotubes and PEDOT:PSS layer. Characterization of Ag-nanowire-embedded PDMS substrate and $WO_3$ nanotube–PEDOT:PSS thin layer. Scanning electron microscope images of (**b**) Ag-nanowire-embedded PDMS, (**c**) $WO_3$ nanotube coated on Ag-nanowire-embedded PDMS, (**d**) cross-section image of a $WO_3$ nanotube, and (**e**) PEDOT:PSS thin layer coated on a $WO_3$ nanotube. (**f**) Transmission electron microscope cross-section image of a PEDOT:PSS thin layer coated on a $WO_3$ nanotube. (**g**) X-ray diffraction results of the $WO_3$ nanotube [85].

In addition to the low-dimensional tungsten oxide structures, the tungsten oxide nanostructures of two-dimensional tungsten oxide nanosheets and three-dimensional tungsten oxide have also been reported, as shown in Figure 8, but the porosity and lifetime of the electrodes composed of both do not reach the low-dimensional level; therefore, the device efficiency and lifetime are low. However, there are some reports in the field of sensors and batteries with low life requirements.

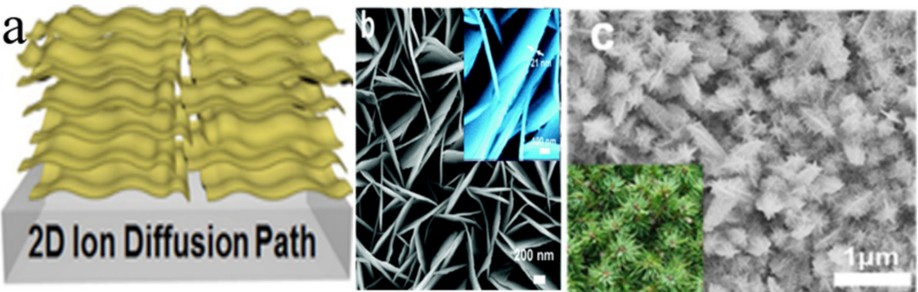

**Figure 8.** Different high-dimensional tungsten oxide nanostructures: (**a**) stacked nanosheet [98], (**b**) nanoarray [99], (**c**) three-dimensional [100].

Table 1 compares many related reports on tungsten-oxide-based ECSCs; mesoporous tungsten oxide electrodes also appear in the ECSC report. Although it does not have the high specific surface area of nanostructure, a mesoporous structure also gives the electrode extremely high porosity; the high-density electrode also gives the device higher energy density, and the cycle stability of the device is higher than that of a loose tungsten oxide electrode. To increase capacitance, carbon materials are widely used. Carbon and graphene oxide, which are widely used in the battery field, are also used in ECSCs. The carbon materials can increase the electric double-layer capacitance of the device. Organic coatings are widely used in tungsten oxide nanostructures, which mainly play a role in two respects: The first is to maintain the stability of the nanostructures and prolong the electrode's life, i.e., preserve the protective layer outside the quantum dots and the organic–inorganic core–shell structure. However, this protective layer will still be damaged in the process of ion entry and exit; thus, in the long run, a new method is needed to improve the stability of the nanostructure. The other is the protective layer between the electrode and the substrate, necessary to meet the needs of flexible devices, e.g., a PEDOT protective layer. It is undeniable that ECSCs are developing in the direction of flexible wearability.

**Table 1.** Relevant tungsten oxide ECSC reports in recent years.

| Time | Film Fabrication Technology | Working Voltage | Specific Capacitance | Coloring Efficiency | Light Modulation Parameters | Base Plate | Article Content |
|---|---|---|---|---|---|---|---|
| 2016 [101] | Solvent-based thermal deposition | −1 V–1 V | | | 20% | Rigid | Using polyaniline (PANI) as the anode, $WO_3$ as the cathode, and polyvinylpyrrolidone (PVP)-LiClO$_4$ as the solid electrolyte, an all-solid-state supercapacitor with electrochromic function was successfully prepared, in which the color change represents the change in energy levels during charging and discharging. |
| 2017 [55] | Spraying | 2.5 V | 406 F/g | 64.8 cm$^2$ C$^{-1}$ | 40% | Flexible | Light-responsive electrochromic supercapacitor with cellulose nanofiber/silver nanowire/reduced graphene oxide/$WO_3$ composite electrode, integrated with a solar sensor to achieve intelligent color change of the device. |
| 2017 [95] | Hydrothermal method | 1.8 V | 459 F/cm$^3$ | | 0.22 a.u | Flexible | $W_{18}O_{49}$ nanowire and carbon nanotube composite electrodes were prepared, using Al-based electrolytes, confirming the promise of Al-based applications in flexible electronic devices. |

**Table 1.** *Cont.*

| Time | Film Fabrication Technology | Working Voltage | Specific Capacitance | Coloring Efficiency | Light Modulation Parameters | Base Plate | Article Content |
|---|---|---|---|---|---|---|---|
| 2017 [102] | Hydrothermal method; pulsed laser deposition | | 15.24 mF/cm$^2$ | 80.6 cm$^2$/C | 68.2% | Flexible | A new flexible electrochromic supercapacitor designed by preparing tungsten trioxide/zinc oxide (WO$_3$/ZnO) nanocomposites on a flexible substrate, which can visually monitor the energy storage level through fast and reversible color change. |
| 2018 [103] | Wet chemical method | | 264 F/g | | | | Tungsten oxide (WO$_3$) nanostructures were synthesized using a wet chemical method and treated using microwave irradiation. The effects of microwaves on the structure, morphology, and performance of WO$_3$ supercapacitors are discussed. |
| 2018 [104] | Solution method | ±2.5 V | 173 F/cm$^3$ | | 63% | Rigid | An electrochromic multifunctional smart glass was assembled by using WO$_3$ crystal nanosheets (a single purely phase-active layer was used for visual and near-infrared (NIR) modulation). The device can be restored to degradation over a period of one year. |
| 2019 [60] | Electrostatic spinning | 1.5 V | 471 F/g | 83.9 cm$^2$/C | 40% | Flexible | Poly(dimethyl siloxane) (PDMS), double-stacked WO$_3$ nanotube electrodes embedded with Au/Ag core–shell nanowires were prepared, and the devices could maintain high performance under stretching and extrusion, which is instructive for work on flexible and stretchable electrochromic supercapacitors. |
| 2020 [105] | Hydrothermal method | 1.0 V | 10.11 mF/cm$^2$ | 608 cm$^2$/C | 40% | Flexible | Construction of P51CA/WO$_3$ composites with PEDOT to form an ECSC. The relationship between the electrochromic coefficient of the material and the electrochromic coefficient of the device was established. |
| 2020 [106] | Magnetron sputtering | ±1.5 V | 147 F/g | | 40% to 50% | Rigid | Preparation of mesoporous WO$_3$ films for combination with solar cells to produce solar-rechargeable electrochromic supercapacitors for self-powered devices |
| 2021 [107] | Hydrothermal method | −0.5 V to 1.0 V | 89.2 mF/cm$^2$ | | 45% | | Asymmetric electrostatic discharge (ESD) based on P5ICN/WO$_3$ was successfully achieved by electrochemical polymerization of WO$_3$-poly(5-cyanoindole) (P5ICN/WO$_3$) hybrids formed by aggregation of nanoclusters, which exhibited good electrochromic and supercapacitive properties. |

**Table 1.** *Cont.*

| Time | Film Fabrication Technology | Working Voltage | Specific Capacitance | Coloring Efficiency | Light Modulation Parameters | Base Plate | Article Content |
|---|---|---|---|---|---|---|---|
| 2021 [108] | Hydrothermal method | −1.7–1.4 V | 21.8 mF/cm$^2$ | 191 cm$^2$/C | 45.8% | Flexible | Three protective layers with different negative ions (i.e., Br, Tf, TFSI) were synthesized as WO$_3$ electrodes. The effects of different counterions and treatment conditions on the cyclic stability and electrochemical properties were investigated. The results show that the introduction of the crosslinked protective layers not only greatly improved the electrochemical stability of the composites, but also led to a significant improvement in the electrochemical properties of the composites, i.e., the phenomenon of activation. |
| 2022 [7] | Spray-film printing | 1.5 V–4.5 V | 102.3 F/cm$^3$ | 93.6 cm$^2$/C | 46.2% | Flexible | Improved flexible amorphous tungsten oxide films by controlling the alkalinity of the solution to inhibit spontaneous crystallization, nucleation, and crystal growth. |

## 3. Conductive Electrodes

Transparent conductive electrodes (TCEs) are a key component of ECSCs. ITO is the most commonly used conductive material in TCEs, with high conductivity and transparency, but pure metal nanowire electrodes are receiving more and more attention based on environmental and material cost considerations. In the field of ECSCs, the use of tungsten oxide nanostructures for flexible ECSCs is increasingly reported, and flexible wearables seem to be a major trend, but the mechanical properties of ITO are weak, and the flexible use will cause the conductivity of the conductive electrodes to decrease. Pure metal nanowire electrodes have the same flexibility advantages as tungsten oxide nanowire electrodes, and their combined use would greatly increase the device performance of flexible ECSCs

Zhang [109] further designed a copper–gold alloy nanonetwork as a flexible transparent electrode on a tungsten oxide nanostructure, as shown in Figure 9. The device assembled from polyaniline and WO$_3$ film by electrochemical synthesis on the metal grid had a coloring efficiency of 153.77 cm$^2$/C and a surface capacitance of 2.29 mF/cm$^2$. The alloy grid's transparent electrodes had a higher conductivity than conventional flexible electrodes; therefore, the devices had a lower operating voltage of 1.0 V, but there was still room for improvement in transparency. Copper and silver nanowires are commonly used as transparent electrode materials, but the low permeability of the materials themselves leads to low light transmission, while air oxidation leads to a decrease in conductivity [110–113].

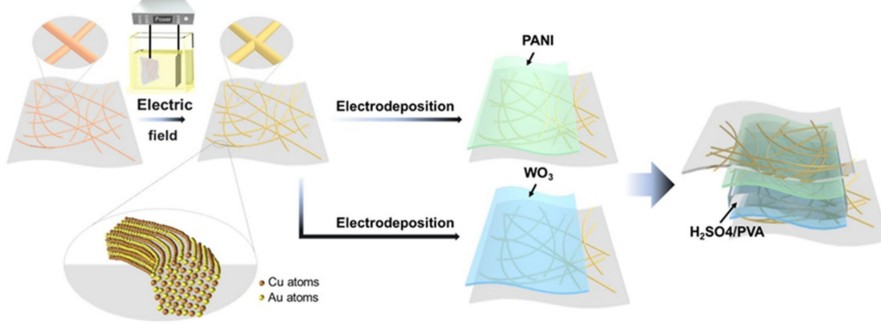

**Figure 9.** Schematic illustration of the fabrication process of an AEES device based on the Cu–Au NW FTEs [109].

## 4. Integrated Uses

The successful integration of EC and SCs illustrates the potential for integration of devices with similar operating parameters, and researchers have attempted to integrate other devices with ECSCs to further enrich the functionality of the devices and broaden the application areas of ECSCs. The basic mode of operation for such integration is usually for the ECSCs to act as a supplementary power supply for the display device to achieve self-powered status. Chang [114] found that the charge and discharge of tungsten-oxide-based electrochromic devices under sunlight is faster. Zhang [115] used dense mesoporous $WO_3$ films to prepare ECSCs that were further integrated with solar cells. The work demonstrated the synergistic performance of solar cell energy storage and electrochromic device light modulation, as shown in Figure 10, effectively reducing the room temperature in a simulated environment and allowing multiple devices to be connected in series to successfully light LEDs. The DSSC module converts solar energy into electrical energy and charges the ECD module under illumination to achieve photochromism, and the electrochromic module can also act as a supercapacitor to store energy and achieve zero energy consumption. PECD has 21 mF/cm$^2$ (114.9 F/g compared to $WO_3$ mass) of electrochemical supercapacitance, with stable mechanical properties and long cycling performance. In addition, the integration of the ECSCs as a low-voltage device with the sensor allows the use of the sensor's electrical signal to change color, and the color changes quickly or slowly to indicate the strength of the sensor signal. The color change can also be used to visually indicate the strength of the sensor signal [116–119].

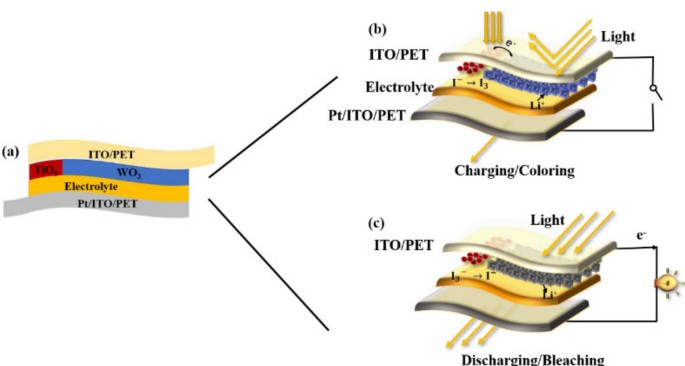

**Figure 10.** Schematic diagrams of the photoelectrochromic device (PECD): (**a**) photo-charging process of the PECD, and (**b**) the discharging process of the PECD (**c**) [115].

## 5. Conclusions and Outlook

Nanometer tungsten oxide electrodes have a large specific surface area, which can meet the electrode requirements of electrochromism and supercapacitors. Tungsten oxide materials with different dimensions have been widely used in electron-capture systems, from zero-dimensional quantum dots to three-dimensional nanowire arrays. In the synthesis part, the use of the protective layer can effectively increase the life of the material; in the application part, the film-forming method is too random, the porosity of the electrode is completely determined by the material itself, the manual control part is less, and the film-forming method is improved so that the arrangement of the nanostructures can be controlled and the performance of the electrode can be further improved. The advantage of flexibility brought by nanometer size is gradually promoting the development of tungsten-oxide-based electrochromic energy storage devices with wearable flexibility, but flexibility also requires higher compatibility between the tungsten oxide electrodes and substrates. The bending performance of the device can be improved by modifying the electrode with organic protective agents and using new transparent alloy electrodes. Finally, for ECSCs as a low-voltage driver, low power consumption is an advantage that cannot be ignored. Combining them with self-powered devices—such as the self-powered ECSCs obtained by

integrating solar cells—would further improve the practicability of the devices, has great application prospects, and is attracting great attention.

**Author Contributions:** Investigation, M.L., J.Z.; resources, X.F.; data curation, X.L.; writing—original draft preparation, M.L., H.Y.; writing—review and editing, M.L., H.Y.; supervision, T.Q., D.L., J.P.; funding acquisition, H.N.; R.Y. All authors have read and agreed to the published version of the manuscript.

**Funding:** This work was supported by the National Natural Science Foundation of China (Grant No.62174057, 62074059 and 22090024), the National Key R&D Program of China (No.2021YFB3600604), the Key-Area Research and Development Program of Guangdong Province (No.2020B010183002), the Guangdong Basic and Applied Basic Research Foundation (Grant No.2020B1515020032), the Special Fund for Science and Technology Innovation Strategy of Guangdong Province in 2021 ("Big Special Project+Task List") Project (No.210908174533730), and the Ji Hua Laboratory Scientific Research Project (X190221TF191).

**Institutional Review Board Statement:** Not applicable.

**Informed Consent Statement:** Not applicable.

**Conflicts of Interest:** The authors declare no conflict of interest.

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
