# Peer review of "Application of Tungsten-Oxide-Based Electrochromic Devices for Supercapacitors"

_asi, doi:10.3390/asi5040060_

Round 1
Reviewer 1 Report
Manuscript ID: asi-1777388
General comments:
In this manuscript (Review) entitled "Application of tungsten oxide-based electrochromic devices for supercapacitors", the authors have concisely described the recent research for application of tungsten oxide based in Electrochromic supercapacitors (ECSCs). This review asserts that WO3-based ECSCs will evolve in the future to provide self-sustaining electricity to fulfill user needs. The work is adequate. The comments may be useful for the improvement of the manuscript. Minor revisions are needed to make the work acceptable.
1. Comment:
In the current state, there are some typographical errors. Like Where M+ is H+, Li+. Therefore, the authors are advised to recheck the whole manuscript for improving the language and structure carefully.
2. Comment:
In order to make more understandable the basic construction of Electrochromic supercapacitors, a general structural diagram of the ECSC must be added in the introduction.
3. Comment:
The future prospectus must be added as separate heading.
4. Comment:
The more reports in recent years relevant to tungsten oxide ECSCS should be included.
5. Comment:
The more possibilities of integrations of WO3-based materials with other materials to ECSCS should be discussed.
6. Comment:
The introduction can be more enriched by adding more relevant literature.
Author Response
I would like to thank the reviewers for their comments. Here is my reply
1.Comment:
In the current state, there are some typographical errors. Like Where M+ is H+, Li+. Therefore, the authors are advised to recheck the whole manuscript for improving the language and structure carefully.
Answer:The typographical errors in the article have been modified accordingly.
2. Comment:
In order to make more understandable the basic construction of Electrochromic supercapacitors, a general structural diagram of the ECSC must be added in the introduction.
Answer: The general structural diagram of the EC has been added in the introduction. According to the description, ECCS and EC have the same structure.
3.Comment:
The future prospectus must be added as separate heading.
Answer; The separate heading “Conclusion and outlook” has been added
4. Comment:
The more reports in recent years relevant to tungsten oxide ECSCS should be included.
Answer: More references have been added. In fact, there are about 50 articles searching keywords with WO3, electrochromic and supercapacitor on web of sci in recent years, and the articles mentioned in this article already include the mainstream reports of WO3 on ECCS, so some similar reports are added to the references, but they are not introduced in detail.
5. Comment:
The more possibilities of integrations of WO3-based materials with other materials to ECSCS should be discussed.
Anwser: Thank you for your suggestion, for Table 1, a description of carbon materials has been added.
6. Comment:
The introduction can be more enriched by adding more relevant literature.
Answer: The relevant literature has been cited in the references

Reviewer 2 Report
The problem of improving supercapacitors is of interest and this review is welcome. The article is well prepared and illustrated, the material is systematized and I believe that the article can be published after minor revision.
1. It would be useful for readers to explain why tungsten oxide was chosen for the review and what are its advantages and disadvantages in comparison with other oxide materials used in supercapacitors (like MnO2, Fe3O4,...).
2. The quality of Figure 2 is poor. It is desirable to improve. Perhaps it makes sense to update this diagram or make your own version, because it is quite old. Although not much has changed since then, some upgrade is still possible.
3. It is desirable to add a conclusion summarizing all of the above and discussing future trends. Also, the structure design of the article could be better thought out.
Author Response
- It would be useful for readers to explain why tungsten oxide was chosen for the review and what are its advantages and disadvantages in comparison with other oxide materials used in supercapacitors (like MnO2, Fe3O4,...).
Answer: Relevant descriptions have been added to the introduction
- The quality of Figure 2 is poor. It is desirable to improve. Perhaps it makes sense to update this diagram or make your own version, because it is quite old. Although not much has changed since then, some upgrade is still possible.
Answer: Thank you for your suggestion, a higher-pixel picture has been used. As you said, this is indeed a classic picture. Even today, it can be seen in various conferences and papers. Its content can fully explain the views expressed in this article, so it seems that it will be more rigorous not to replace..
- It is desirable to add a conclusion summarizing all of the above and discussing future trends. Also, the structure design of the article could be better thought out
Answer: The separate heading “Conclusion and outlook” has been added
